# Fractal Analysis of Fuel Nozzle Surface Morphology Based on the 3D-Sandbox Method

**DOI:** 10.3390/mi14050904

**Published:** 2023-04-23

**Authors:** Yeni Li, Liang Hou, Yun Chen

**Affiliations:** 1Pen-Tung Sah Institute of Micro-Nano Science and Technology, Xiamen University, Xiamen 361102, China; liyn@xmut.edu.cn (Y.L.); hliang@xmu.edu.cn (L.H.); 2School of Mechanical and Automotive Engineering, Xiamen University of Technology, Xiamen 361024, China

**Keywords:** fuel nozzle, surface morphology, fractal dimension, 3-D sandbox counting method

## Abstract

The dual oil circuit centrifugal fuel nozzle is made of martensitic stainless steel, which has complex morphological characteristics. The surface roughness characteristics of the fuel nozzle directly affect the degree of fuel atomization and the spray cone angle. The surface characterization of the fuel nozzle is investigated by the fractal analysis method. A sequence of images of an unheated treatment fuel nozzle and a heated treatment fuel nozzle are captured by the super-depth digital camera. The 3-D point cloud of the fuel nozzle is acquired by the shape from focus technique, and its three-dimensional (3-D) fractal dimensions are calculated and analyzed by the 3-D sandbox counting method. The proposed method can characterize the surface morphology well, including the standard metal processing surface and the fuel nozzle surface, and the experiments show that the 3-D surface fractal dimension is positively correlated with the surface roughness parameter. The 3-D surface fractal dimensions of the unheated treatment fuel nozzle were 2.6281, 2.8697, and 2.7620, compared with the heated treatment fuel nozzles dimensions of 2.3021, 2.5322, and 2.3327. Thus, the 3-D surface fractal dimension value of the unheated treatment is larger than that of the heated treatment and is sensitive to surface defects. This study indicates that the 3-D sandbox counting fractal dimension method is an effective method to evaluate the fuel nozzle surface and other metal processing surfaces.

## 1. Introduction

As one of the key components of fuel atomization in an aero-engine combustion chambers, the surface morphology of the fuel nozzle directly determines the effect and uniformity of fuel atomization. There are three types of commonly used fuel nozzles: direct injection, dual-orifice pressure-swirl atomizer, and pneumatic atomization. Among them, the dual-orifice pressure-swirl atomizer has higher atomization quality for wider application, but its auxiliary part has a small geometric size, a complex structure, and great processing difficulty. The surface topography is very important to the function of the aero-engine, making an urgent need for a comprehensive evaluation of the inner surface topography of the fuel nozzle.

In traditional evaluation methods of fuel nozzles, a number of statistical parameters are used to depict various aspects of the surface roughness. Ding et al. [1] proposed an approximate algebraic solution for the discharge coefficient of the sonic nozzle with surface roughness; the method gives an excellent tool to further investigate the effect of surface roughness. Li et al. [2] evaluated the internal wall surface characteristics of fuel nozzle micro-orifices using the synchrotron radiation X-ray micro-CT technology. This new technology has been improved progressively, with an already achieved spatial resolution of 1.85 μm. Pan et al. [3] conducted experimental measurements and numerical calculations to investigate the influence of surface roughness on the particle deposition distribution around a multi-slot cabin supply air nozzle. However, most of the statistical parameters, such as *Ra*, *Ry* and *Rq*, strongly depend on actual measured conditions, including the sampling, the scan lengths, the instrumental resolution, and so on [4].

In addition, these conventional parameters are not enough to characterize the surface morphology, and some spatial texture can be lost. Therefore, new parameters to characterize the surface roughness are necessary. Fractal was initiated by Mandelbrot to describe the degree of irregularity of a machined surface [5]. Over a period of time, it has been used as a very effective tool in the topographical characterization of complex surfaces. Topography parameters derived from a fractal model are believed to be intrinsic characteristics of surfaces and considered to be independent of scale, sampling, and filtering.

Methods employed to calculate fractal dimension include the box counting method, the variation method, the power spectral density function method, the structure function method, the root mean square method, the wavelet analysis method, etc. There were several studied that evaluated the surface topography of machined surfaces using fractal methods based on box counting. Each method has its own application scope. In an earlier work, Lopez et al. [6] studied the fractal characterization of the sandblasted surface. Their results show that the numerical determination of LH exponents is adapted. Wu et al. [7] developed a stable cubic covering method (SCCM) and stable differential cubic covering method (SDCCM) based on the box-counting method, and found that the new methods are more stable than previous ones on a real sandstone fracture surface. Zuo et al. [8] proposed a fractal dimension and characteristic roughness models for the turned surface of carbon steels. Experiments are carried out on a CNC lathe for six carbon steel material, and show that material hardness and spindle speed have the same influence on the fractal dimension. Woraratpanya et al. [9] present a method called the triangle box counting method to avoid the demerits of the BC method.

The box counting method and the differential box counting method are widely used in fractal dimension measurement. However, there are some obvious flaws in using the box counting method. Thus, it should be improved to solve those flaws. Zhao et al. [10] improved the fractal contact model to obtain the fractal contact compliance, and the numerical simulation results show that the contact stiffness increases with the increment of the fractal dimension and the reduction of the fractal scale characteristic coefficient. Arnold et al. [11] investigated the quantification and comparison of roughness data obtained by 2-D and 3-D methods, evaluating the variations of results between the measurement methods and formulating a critical analysis regarding the quality of the information obtained with each method. Song et al. [12] studied the fractal theory to calculate the fractal dimension to evaluate the morphological characteristics of aggregates. Their results show that fractal dimensions measured by the box-counting and sandbox methods were well-correlated with indicators of high-temperature ability. Later on, a more accurate and reasonably fast algorithm was developed. Panigraphy et al. [13] introduced three improved DBC methods using three box heights based on the eigenvalue, kurtosis, and skewness of an image. Silva et al. [14] proposed a mathematical analysis of the model, which is accomplished here by employing techniques from statistics and combinatorics, confirming the potential of a combination of box counting, fractal dimension, and statistics. Cimen et al. [15] studied the fractal size-based tissue analysis method, which is calculated by the box counting method, and showed that the method yields promising and optimistic results.

A literature review indicates some problems. For example, different papers reached the opposite conclusion about the relationship between traditional surface roughness and fractal dimension on the machined surfaces; some reported positive correlations, while others reported a negative correlation [7]. For different materials, the surface fractal properties were significantly different from the 2-D or 3-D profile data [16]. Moreover, the inner hole surface of fuel nozzle is difficult to detect by traditional measuring instruments, and the uniformity of surface morphology is not as good with manual grinding. The inner surfaces have complicated anisotropic properties. These anisotropic properties of metal grinding surfaces are difficult to detect by 2-D roughness methods. Thus, it becomes important to find possible quantitative ways of characterizing anisotropy. However, by now, the machined inner surface of the fuel nozzle is barely evaluated by fractal dimension, so it is reasonable to pay great attention to investigating their fractal properties. This paper explores the 3-D sandbox counting method to characterize the 3-D surface morphology of fuel nozzle inner surfaces. The 3-D fractal properties of heated and unheated treatment surfaces are calculated and analyzed by this fractal method in order to comprehensively evaluate and optimize the metal surface morphology.

This paper is organized as follows. Firstly, Section 2 presents the 3-D point cloud extraction principle of the fuel nozzle, including the most important influencing factors of the shape from the focus technique. Then, the specific implementation process of the 3-D sandbox algorithm is shown in Section 3. Subsequently, in Section 4, the experiment results showed that the improved 3-D sandbox method can expose the surface morphology. Finally, the major conclusions are drawn in Section 5.

## 2. 3D Point Cloud Extraction Principle of the Fuel Nozzle

Shape from focus (SFF) is a passive optical method utilizing 2-D focus information as a cue to reconstruct a 3-D object/scene [17]. The concept of SFF was first proposed by Nayar [18]. SFF techniques are usually performed in the following steps [19]. As shown in Figure 1. The first step is to acquire the sequence images. When the camera moves from bottom to top, some points of the image will be focused and others will be defocused. During the movement, the focus area will gradually change. In the second step, the focus measure (FM) operator is applied to calculate the focus quality of the image pixel blocks. At the third step, the depth of the best focus position will be extracted by evaluating the sharpness of the image pixel blocks. After traversing all the pixel blocks, the depth information of the entire image can be obtained. The best focused position will be searched, and the frame number will be stored.

The shape from focus algorithm is used to extract the three-dimensional depth point cloud data of the fuel nozzle. Different sharpness evaluation operators and window sizes of focus evaluation will directly affect the quality of focus evaluation, so it is very important to find the best focus evaluation operator and window size [20]. A small window size can preserve depth discontinuities but increase the sensitivity to noise. Meanwhile, a large window size performs better for noisy images, but at the cost of blurring sharp edges. Thus, a window size of 24×24 is accepted and proved to be the best one [21].

In order to find the best-focused position of the fuel nozzle, there are four operators used to evaluate the sharpness performance. The operators are as follows: absolute gradient, Laplacian, Tenengrad, and Brenner. Moreover, different window sizes are used to evaluate the optimal size. The absolute gradient [22], which is also known as the sum modulus difference (SMD), is the sum of the absolute horizontal and vertical image gradients. The Tenengrad focus measure is proposed by Tenenbaum and Schlag [23]. It is a focus measure similar to the squared gradient, which uses the Sobel operator. The Laplacian focus measure is proposed by the sum of the absolute values of the second-order derivatives [24]. The Brenner focus measure developed by Santos et al. [25] was devised for automated microscopy; when the focus increases, the differences between the image pixels and their neighbor’s increase. An illustration of the focus measure curve obtained by different operators of the fuel nozzle image sequence is shown in Figure 2a,c. Obviously, the best focus measure operator was the Tenengrad operator, because of its robustness and best peak value. As shown in Figure 2b, the window sizes of points A and B are both 24×24, and the best focus frame is the frame 31.

## 3. Fractal Characterization of Fuel Nozzle

Fractal theory was introduced by Mandelbrot [5] to describe the degree of irregularity in objects. Morphology evaluation based on fractal theory is believed to be an intrinsic characteristic of surfaces and is considered to be independent of scale sampling and filtering [4]. Fractal geometry has many advantages in studying the irregular surface due to the machining trace. Thus, fractal parameters instead of the traditional ones are shown to describe the fuel nozzle surface morphology.

For different kinds of research, the surface fractal properties can be detected by 2-D or 3-D methods but show opposite conclusions between the fractal dimension and surface roughness [26]. The fuel nozzle inner hole surface is processed by manual grinding. For some pieces not treated with heat, the roughness of the inner hole surface is too large to achieve an optimal atomization effect. The machined fuel nozzle surfaces are barely evaluated by the fractal method; thus, it is reasonable to pay great attention to investigating their fractal properties. The fractal characteristics of fuel nozzle surfaces are calculated and analyzed by the 3-D sandbox counting fractal method in order to comprehensively evaluate and optimize the topographic structures.

Fuel nozzle used to turn, debur, drill, grind, and polish after heat treatment, which is shown in Figure 3. Through a series of processing technologies, the surface of the fuel nozzle has obvious fractal characteristics. Due to technical limitations, the debugging and intermediate testing after secondary clamping have not been implemented, and the cleaning and final testing of the nozzle after heat treatment have not been implemented. A comparison and analysis of the fractal dimension based on the 3-D sandbox method between the intermediate detection and the final detection was accomplished.

### 3.1. Box-Counting and Sandbox-Counting Methods

To calculate the fractal dimension of the fuel nozzle inner hole surface, two methods, such as the box-counting method [27] and the sandbox-counting method [28], can be used.

The box counting method is one of the simplest and most popular methods for estimating fractal dimensions. As shown in Figure 4a, there are some squares coved on the image; the total number of squares is changed by the different size of r. In other words, the number of squares *Nr* required to cover the fractal pattern for different lengths and sizes of r is counted.

The sandbox method is a variant of the box counting method. In the sandbox method, a square of size r is increased until it equals half of the image size. The number of pixels, *Nr*, can be evaluated by counting all of the points in the square shown in Figure 4b. Thus, *Nr* is obtained for different square of sizes r. Fractal dimension *D* is the exponent that expresses the scaling of the bulk with its size.

As an important parameter of fractals, the fractal dimension *D* can be used to measure irregularities and surface roughness. *D* is a non-integer parameter that is compared with the topological dimension of conventional geometry such as *Ra*, *Rs* [29].

A fractal characteristic does not necessarily contain a distinguishable replication of itself but maintains a similar irregularity at different scales. The inner surface of the fuel nozzle has an irregular geometric machined form, so it can be fragmented by means of smaller objects and be similar to the original nozzle. The fractal dimension (*D*) can be derived from Equation (1) using the fractal computation method [30]. When the inner and outlet surfaces of the fuel nozzle are covered by boxes (or other shapes) of size *r*, *D* obtained by the logarithm of the number of boxes (or other shape) against the logarithm of the relative box size *r*.
(1)$$D=-\frac{\log(N)}{\log(1/r)}=\frac{\log(N)}{\log(r)}$$
where *N* represents the number of boxes (or other shapes) of different scales by ratio *r*. Researchers have evaluated the fractal dimension using several techniques, with results that are reported being different or even opposite to the findings of other studies; thus far, no one has proven the superiority of their method [31]. Thus, for a series of different lengths of *r*, applying the least squares method to simulate the linear functions of *log*(*N*) and *log*(*r*) gives the slope of the linear function as equal to the sandbox-counting dimension *D* [32]. We should ensure that the radius size of *r* is twice as large as the interval between the points.

Both the box-counting method and the sandbox-counting method follow the below steps:

First, different sizes of *r* is considered to compute the fractal dimension value of the object.

Second, the regression line is plotted according to the different *r* and corresponding *N*.

Third, *D* can be computed using the slope of the regression line.

### 3.2. Improved 3-D Sandbox-Counting Method

The 2-D and 3-D box-counting dimensions are widely used in fractal estimation because of their ease of mathematical calculation and empirical estimation [33]. However, the box-counting method cannot completely cover the adjacent boundary; the calculated fractal dimension will lead to under-counting problems that affect the accuracy of the results [34], which can lead to considerable errors. The improved sandbox counting method is proposed because of its high computational efficiency and calculation accuracy. In this study, a new 3-D sandbox-counting method is used to analyze inner surface topography. Figure 5 shows an illustration of fractal analysis using the 3-D sandbox-counting method.

Therefore, the quantitative description of the topography of the machining object’s surface is very important [33]. Three-dimensional (3-D) research studies have been performed on the surface of the fuel nozzle.

The above-mentioned problems indicate some defects in the existing box and sandbox methodologies; we optimized the formula and developed it into a 3-D point cloud fractal method.

The sandbox method measured the 2-D image or 3-D point cloud by covering a square (circle) or square (cylinder) with a size of r(r>1). Based on the gray value of the 2-D image pixel or the actual *z*-axis height value of the 3-D point cloud, calculate the total number of the 2-D pixel values or points of clouds *N (r)*. Then gradually increase the size of *r* to obtain the total number of pixels or point clouds corresponding to different radii r(r>1).

Some research studies converted the 2-D image into a 3-D image, where the *x*-axis corresponds to the horizontal direction of the image, the *y*-axis corresponds vertical direction of the image and the *z*-axis corresponds to the gray level of the image [34]. The 3-D sandbox method is presented to compute the point cloud fractal dimension in order to improve the computation accuracy.

The sandbox fractal method is based on box-counting and has been applied in many fields, but there is no research on the characterization of fuel nozzle surface morphology [35]. Since the inner hole of the fuel nozzle is a cone and the cross section is a circle, the cylinder is used to cover the 3-D point cloud of the fuel nozzle. The fractal dimension value of the inner surface or outlet surface is computed from different point clouds.

The 3-D point cloud was extracted from the sequence images based on shape using focus technology. Each image of 5496×3672 pixels, was used in the experiment. The best window for sharpness evaluation is 24×24. Then the number of points in the *x*-axis and y-axis of the three-dimensional point cloud is 229×153, and the height is the actual number of frames. After calibration, the real point cloud data is obtained. The *z*-axis point cloud is filtered according to the point cloud distribution and coordinate mean value to obtain a reduced point cloud. Then 3-D sandbox counting is used to calculate the fractal dimension, which is shown in Figure 6. The central hole of the fuel nozzle surface is taken as the growth center to calculate the fractal dimension value. Specific steps are described as follows:

Step 1: First, the initial point cloud was extracted by the shape from focus technology, and the number of points is 229×153.

Step 2: The calibrated 3-D point cloud is filtered and the point cloud is filtered and reduced by radius filtering to obtain the simplifying point cloud.

Step 3: Set the center of the point cloud as the growth seed; the initial radius of the cube is 10 × 10 × 10. The number of points in the square is denoted by *N (r)*.

Step 4: Gradually, the size of the cube is increased by 10 points every time. The height of the *z*-axis remains the same until the square size reaches half of the point cloud width and height. The number of three-dimensional point clouds in the cube are recorded as *N* (10), *N* (20), *N* (30), *N* (40), *N* (50) … *N (r)*.

Step 5: The logarithm of the number of point clouds *N (r)* in the square and the square size *r* for each time are taken, and the least squares method was used to conduct linear regression on the distribution points, and obtain the slope of D, which is the three-dimensional fractal dimension of this 3-D region.

## 4. Experiment

The sequence image acquisition system was designed to obtain the 3-D point cloud of the fuel nozzle based on the shape obtained from focus technology. All experiments were run on a PC with an Intel Core i7 2.9 GHz processor and 16 GB of RAM. They were performed on the three-axis platform. The camera fixed on the platform was MER-2000-19U3C (5496 × 3672 resolution), and the microscope OPTEM204310 was used in the experiment. The *z*-axis was controlled by the stepper motor and driver. The motion position of the *z*-axis was recorded by an encoder. The computer software controlled the position of the lens through a stepper motor driver to obtain multi-focus images. Thus, a sequence of 156 images of the fuel nozzle, each of 5496×3672 pixels, was used in the experiment. The objective lens of the microscope was 1.5× and the number of sequence images was 156. The software implementation and the validation were done in LabVIEW and MATLAB on sequence images of the fuel nozzle.

### 4.1. 3D Fractal Calculation of a Standard Sample

The all-in-focus turning surface image is shown in Figure 7a. The 3-D point cloud of different criterion surface morphologies is shown in Figure 7b. The four criteria for turning surface roughness Ra are 6.30 μm, 3.20 μm, 1.60 μm, and 0.80 μm, respectively. The fractal dimensions based on the 3-D sandbox method are 2.7909, 2.3160, 2.0782, and 2.1181, respectively, as shown in Figure 7c. As the roughness decreases, the fractal dimension decreases, but there is no strict linear relationship. Assuming that the 3-D sandbox of the *z*-axis is increased by 2, r=2 to r=20, boxes were used to cover the turning workpiece which surface roughness is Ra=6.30um. In Figure 7a, the number of the boxes were *N* (2) = 16, *N* (4) = 112, *N* (6) =349, *N* (8) = 836, *N* (10) = 1435, *N* (12) = 1543, *N* (14) = 4062, *N* (16) = 5708, *N* (18) = 7194, *N* (20) = 9119. Then the scatter plot of *lnN(r)* versus *ln(1/r)* is presented based on the boxes size r used in the previous step and the resultant *N (r)* is shown in Figure 7c, where the box size of the *z*-axis ranges from 2 to 20. The least squares method is used for line fitting to obtain the slope of the fitted line. As shown in Figure 7c, the fractal dimension value obtained from the slope of fitted line is D=2.7909. The results showed that when the roughness is small, the fractal dimension becomes smaller. The accuracy of the image acquisition will affect the calculation of fractal dimensions. When the image has defects or reflections, the point cloud has more noise, and the fractal dimension will become larger. Thus, the third turning image which is shown in Figure 7a, its Ra is 1.60 μm, but the fractal dimension based on the 3-D sandbox method is 2.0782. The fourth turning image’s Ra is 0.80 μm, but the fractal dimension is 2.1181, which is larger than the third image because of the noise points.

The all-in-focus polished surface image is shown in the Figure 8a. The 3-D point cloud of different criterion surface morphologies is shown in Figure 8b. The four criteria for polished surface roughness Ra are 0.40 μm, 0.20 μm, 0.10 μm, and 0.05 μm, respectively. The fractal dimensions based on the 3-D sandbox counting method are 2.7354, 2.5241, 2.3435, and 2.1070, respectively, as shown in Figure 8c. The result show that when the roughness increases the fractal dimension also increases. With the decrease in roughness, the more noise points appear in the 3-D point cloud. The 3-D point cloud of different polished surface morphologies is shown in Figure 8b. Assuming that the 3-D sandbox of the *z*-axis is increased by 2, *r* = 2 to *r* = 20, boxes were used to nearly cover the object with the surface roughness as *Ra* = 0.40 μm in Figure 8a. The number of the boxes were *N* (2) = 15, *N* (4) = 230, *N* (6) = 482, *N* (8) = 1128, *N* (10) = 1820, *N* (12) = 3047, *N* (14) = 4907, *N* (16) = 6418, *N* (18) = 8131, *N* (20) = 10,149. Then the scatter plot of *lnN*(*r*) versus *ln*(1/*r*) is presented based on the box size *r* used in the previous step and the resultant *N* (*r*), where the box size of the *z*-axis ranges from 2 to 20. The least squares method is used for line fitting to obtain the slope of the fitted line. As shown in Figure 8c, the fractal dimension value obtained from the slope of the fitted line is D=2.7354.

The smoother the surface is, the darker the image surface is when the sequence image is used to synthesize the all-in focus image, so noise leads to some deviation error. On the other hand, the smoother the image is, the more obvious the impact of defect points. When the texture surface has flaws, more point cloud noise will be extracted. The image acquisition of polishing and grinding also causes deviations in the calculation of fractal dimensions due to the appearance of noise. The fractal dimension reflects the roughness of the actual point cloud. If there are many noise points, it is recognized as the roughness of the point cloud, not the real texture roughness, so it does not reflect the real surface.

The 3-D sandbox fractal dimension showed a positive correlation with the roughness Ra of turning and polished surfaces. However, there is no obvious rule between different materials. In the morphology fractal of turning materials, with the surface roughness ranging from 6.3 μm to 0.80 μm, the three-dimensional fractal dimension is 2.79091 to 2.0782. The roughness of the polished surface is relatively small, ranging from 0.40 μm to 0.05 μm. As the roughness decreases, the fractal dimension value also decreases. The three-dimensional fractal value D of the polished image surface ranges from 2.7354 to 2.1070. However, the fractal dimension and surface roughness between turning and polishing images do not show obvious regularity.

### 4.2. 3-D Fractal Calculation of Fuel Nozzle

The 3-D point cloud of the unheated treatment fuel nozzle is extracted from the sequence images based on its shape using focus technology. The number of points is 229×153. The point cloud center is (115, 75). Based on the center point, the initial cube based on the 3-D sandbox method is obtained. The *x*-axis range is 110–120, *y*-axis is 70–80, and the *z*-axis is the actual point of 20 to 30. Then the cube is expanded by 10 points every time. The *x*-axis of the second cube is 105–125, the *y*-axis is 65–85, and the *z*-axis is 20–40. The *x*-axis of the third cube is 100–130, the *y*-axis is 60–90, and the *z*-axis is 20–50. According to this rule, the point cloud is covered by increasing the cube until the last box reaches the edge of the point cloud. The length of cube size of the *x*-axis is 55–175, the *y*-axis is 15–135, and the *z*-axis is 20–140. The logarithm of the number of point clouds contained in each box and the logarithm of the side length of the box are linearly fitted by the least squares method, and the slope of the fitted line is the fractal dimension value of the point cloud. If the side length of the cube is r and the number of point clouds is *N* (*r*), then *N* (10) = 2, *N* (20) = 27, *N* (30) = 313, *N* (40) = 856, *N* (50) = 1651, *N* (60) = 2684, *N* (70) = 3963, *N* (80) = 5405, *N* (90) = 6998, *N* (100) = 8787, *N* (110) = 10,817, and *N* (120) = 13,205. The first two points are at the outlet hole position and do not belong to the inner surface of the nozzle, so they are deleted. The other ten points are fitted by the minimum binary method, and the slope is 2.6281.The inner hole surface fractal dimension of the No. 1 nozzle is D=2.6281, which is shown in Figure 9a. The fractal dimension of the No. 2 nozzle is D=2.8697, which is shown in Figure 9b. The fractal dimension of the No. 3 nozzle is D=2.7620, which is shown in Figure 9c.

The 3-D point cloud of the heat treatment fuel nozzle is extracted from the sequence images based on its shape using focus technology. The number of points is 229×153. The point cloud center is (110, 75). Based on the center point, the initial cube based on the 3-D sandbox method is obtained. The *x*-axis range is 105–115, *y*-axis is 70–80, and the *z*-axis is the actual point of the hole, *z*-axis range of the initial box is 12 to 22. Then, the cube is expanded by 10 points every time. The *x*-axis of the second cube is 100–120, the range of the *y*-axis is 65–85, and the *z*-axis is 12–32. The third cube is 95–125, the *y*-axis is 60–90, and the *z*-axis is 12–42. According to this rule, the point cloud is covered by increasing cube until the last box reaches the edge of the point cloud. The length of cube size of the *x*-axis is 60–160, *y*-axis is 25–125, *z*-axis is 12–112. The logarithm of the number of point clouds contained in each cube and the logarithm of the side length of the cube are linearly fitted by the least squares method, and the slope of the fitted line is the fractal dimension value of the point cloud. If the side length of the cube is r and the number of point clouds is *N* (*r*). Thus, *N* (10) = 73, *N* (20) = 195, *N* (30) = 550, *N* (40) = 1183, *N* (50) = 1949, *N* (60) = 3018, *N* (70) = 4342, *N* (80) = 5870, *N* (90) = 7592, *N* (100) = 9489, *N* (110) = 11,607, and *N* (120) = 13368. The first two points are at the outlet hole position and do not belong to the inner surface of the nozzle, so they are deleted. The other ten points are fitted by the minimum binary method, and the slope is 2.3021. So, the inner hole surface fractal dimension of the No. 4 nozzle is D=2.3021, which is shown in Figure 10a. The fractal dimension of the No. 5 nozzle is D=2.5322, which is shown in Figure 10b. The fractal dimension of the No. 6 nozzle is D=2.3327, which is shown in Figure 10c.

Figure 11 shows the fractal dimension fitting of the inner hole morphology of the heated treatment of the No. 4 fuel nozzle. In Figure 11a, there were 12 original fitting points, but in Figure 11b, there were 10 fitting points. The two points with red circle in Figure 11a had been deleted because there was deviation. Since the first two points are in the position of the nozzle outlet central hole, they belong to the detection platform, not the part of the nozzle inner hole surface. So, the fitting points have a deviation, and the fractal dimension value is 2.2147. The two points are deleted, and the corrected fractal dimension is 2.3021. In the inspection of the inner hole of the fuel nozzle, the 3-D sandbox method is used to calculate the fractal dimension of the point cloud. The results showed that the fractal dimension of the inner hole after heat treatment is smaller than that of the inner hole after unheated treatment. The fractal dimension of the inner hole surface with defects is larger than that without defects.

## 5. Conclusions

According to the 3-D sandbox counting models, the relationship between roughness and the fractal dimension of the fuel nozzle surface can be predicted. The proposed experimental and statistical approaches provide reliable results to improve surface processing technology. Although the results may be different, the method can be efficiently extended to evaluate other materials’ surface morphology. A simple and reliable procedure for the determination of fractal surface parameters was presented. The 3-D sandbox fractal method is explored for the possible evaluation of the fuel nozzle surface topography. The effort of computation is small compared with other methods of fractal analysis.

The experiment results show that standard metal machine surfaces and the fuel nozzle surfaces exhibit good fractal morphology. From analyzing the results using the 3-D sandbox counting method, the following conclusions can be obtained:

(1) The 3-D surface fractal dimension showed a positive correlation with the roughness Ra of fuel nozzle surfaces. The fractal dimension is sensitive to the surface defects and noise of the point cloud. A large D value means a worse surface quality.

(2) The evaluation of standard metal surfaces showed that the surface quality can be implied from the fractal dimensions. However, the fraction dimension of the smooth surface has a deviation error because of the noise in the point cloud.

(3) The inner surface of the fuel nozzle with heated treatment has better quality than the unheated treatment surfaces. The fractal dimension of heated treatment surfaces is smaller than that of unheated treatment surfaces.

The 3-D sandbox counting method relies on the quality of sequence images, which directly leads to less error in the fractal dimension counting. Thus, it is necessary to conduct research to identify the best measure operator to extract the accuracy 3-D point cloud with less noise. In addition, it is also necessary to further study the correlation between the fractal dimension value of the inner hole and the specific processing technology so as to achieve the goal of s smoother inner surface by improving the technology.

## Figures and Tables

**Figure 1 micromachines-14-00904-f001:**
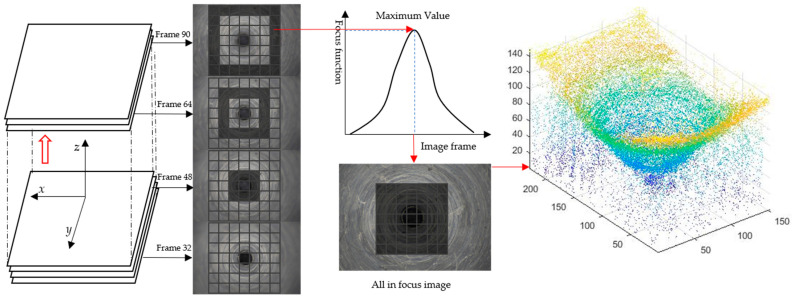
3-D point cloud extraction principle based on shape from focus.

**Figure 2 micromachines-14-00904-f002:**
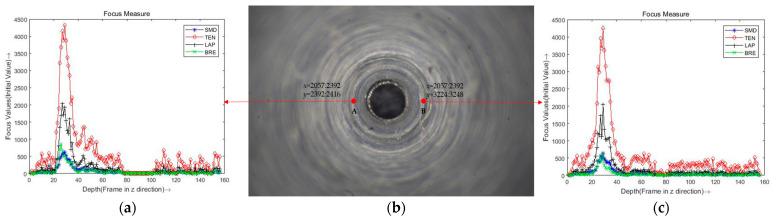
An illustration of the focus measure curve obtained by different operators of the fuel nozzle image sequence. (**a**) A pixel block with a window size of 24. (**b**) The image of frame 29. (**c**) B pixel block with a window size of 24.

**Figure 3 micromachines-14-00904-f003:**
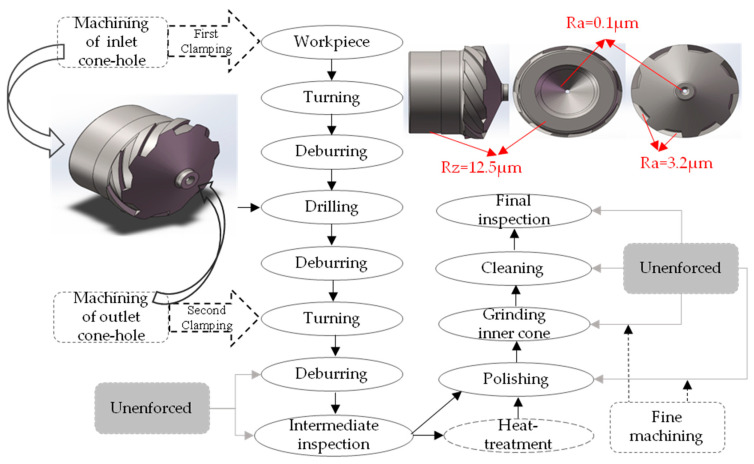
Machining process and roughness parameter of the fuel nozzle.

**Figure 4 micromachines-14-00904-f004:**
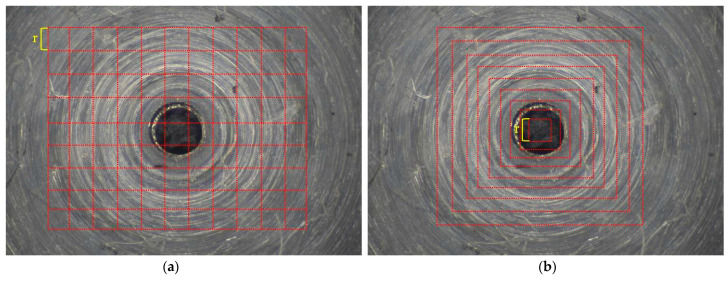
2-D Fractal dimension computational procedures (**a**) Box-counting. (**b**) Sandbox-counting.

**Figure 5 micromachines-14-00904-f005:**
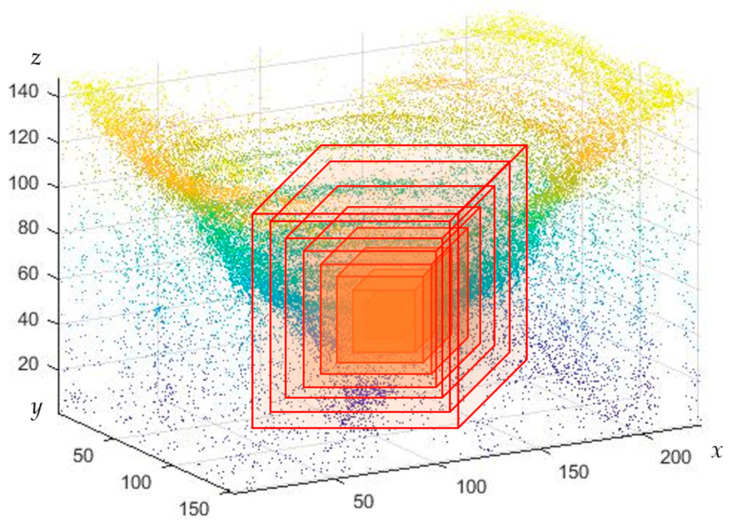
The model of the 3D-sandbox method.

**Figure 6 micromachines-14-00904-f006:**
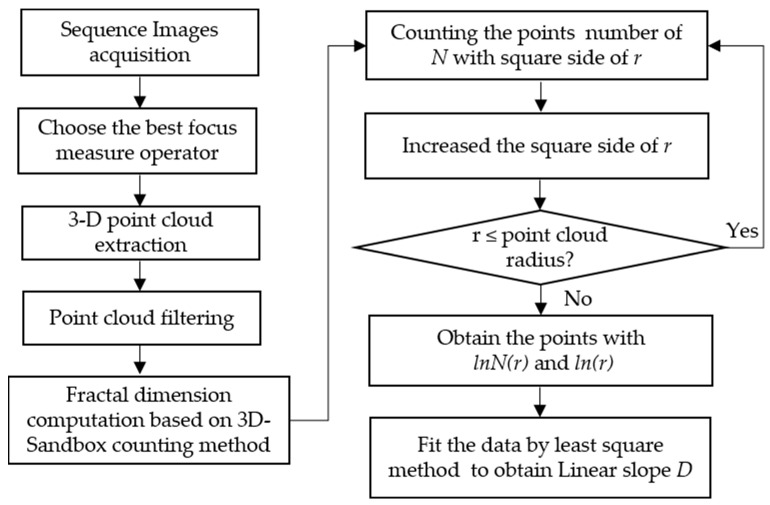
The diagram of the 3-D sandbox counting method.

**Figure 7 micromachines-14-00904-f007:**
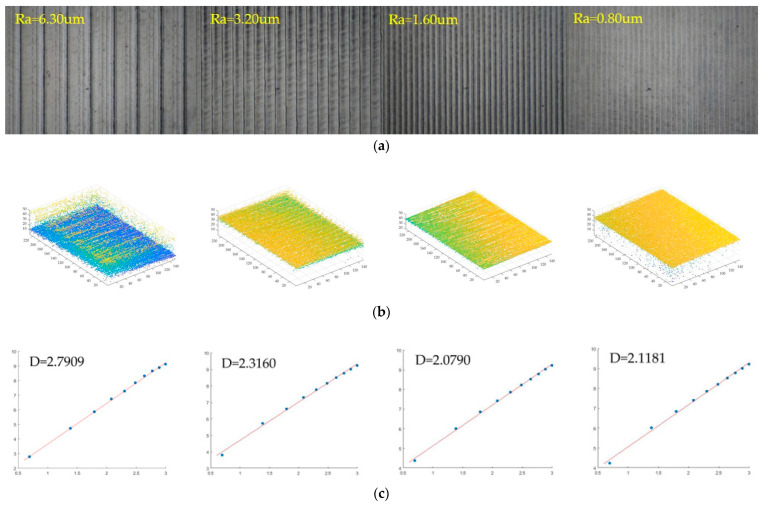
Fractal dimension calculation on the turning surface. (**a**) All in focus; (**b**) 3-D point cloud; (**c**) Least squares fitting curve.

**Figure 8 micromachines-14-00904-f008:**
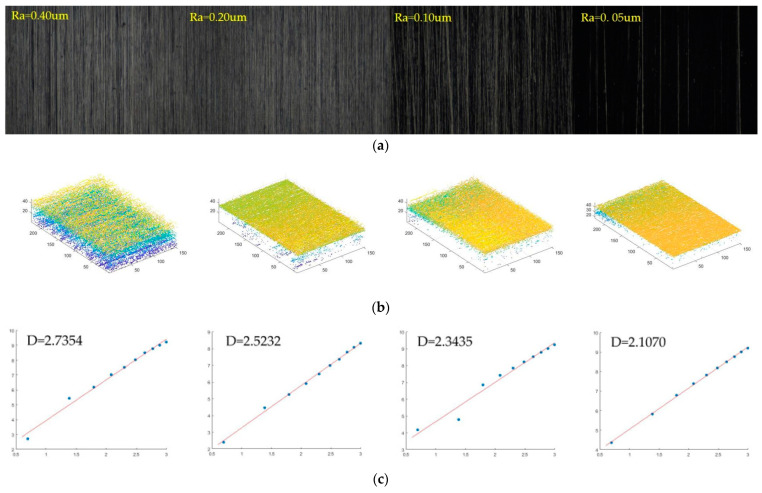
Fractal dimension calculation on a polished surface. (**a**) All in focus image; (**b**) 3D point-cloud; (**c**) Least squares fitting curve.

**Figure 9 micromachines-14-00904-f009:**
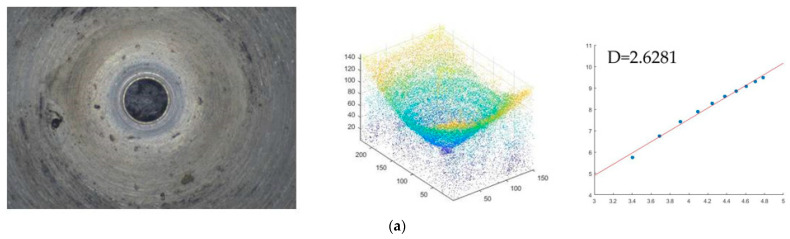
Fractal dimension of the point cloud of the unheated treatment fuel nozzle (**a**) No. 1 fuel nozzle. (**b**) No. 2 fuel nozzle (**c**) No. 3 fuel nozzle.

**Figure 10 micromachines-14-00904-f010:**
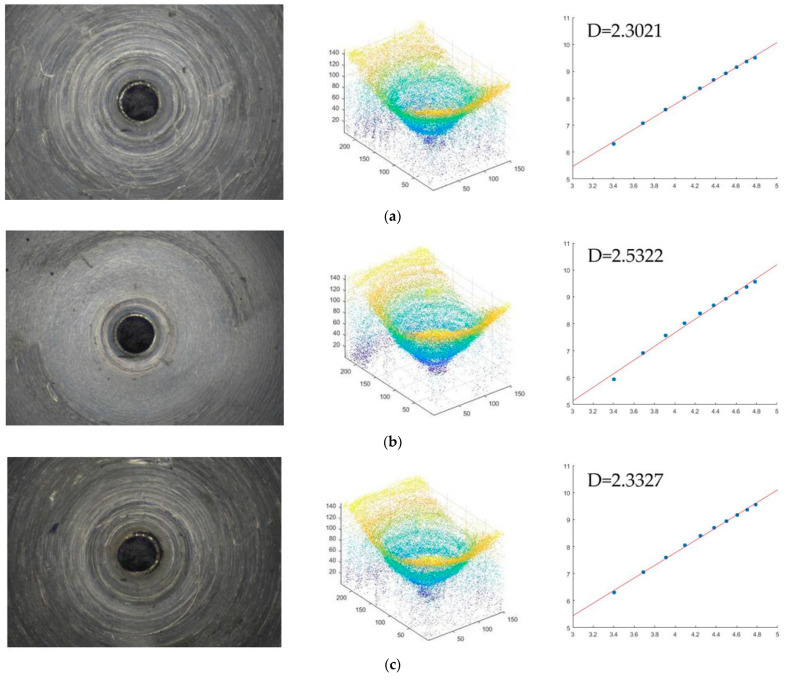
Fractal dimension of the point cloud of the heated treatment fuel nozzle (**a**) No. 4 fuel nozzle. (**b**) No. 5 fuel nozzle (**c**) No. 6 fuel nozzle.

**Figure 11 micromachines-14-00904-f011:**
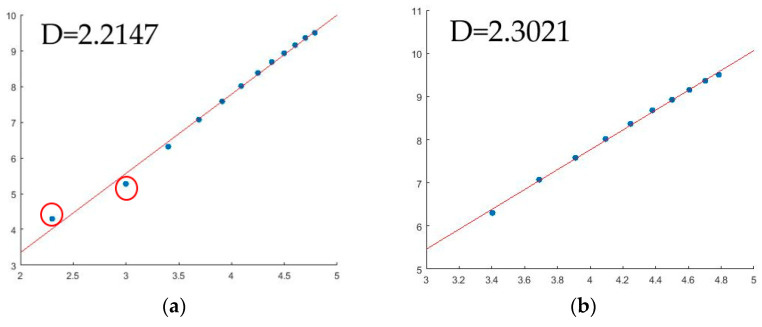
3-D sandbox counting analysis of the No. 4 fuel nozzle surface. (**a**) The original fractal dimension results of the 3-D point cloud. (**b**) The corrected fractal dimension results of the 3-D point cloud.

## Data Availability

Not applicable.

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
