# Peer review of "Fractal Analysis of Fuel Nozzle Surface Morphology Based on the 3D-Sandbox Method"

_micromachines, 2023, doi:10.3390/mi14050904_

Round 1

Reviewer 1 Report

The topic is very interesting, and the work contains a lot of analysis which is worth being communicated to the general public. Nevertheless, as usual, establishing a correlation between parameters resulting from the analysis and really useful surface properties is a very hard task, which has not been faced up clearly in this work.

I recommend minor revisions according to the following detailed directions.

1.      Abstract should contain more quantitative information.

2.      Introduction may be improved through the inclusion of few more recent previous works done on fuel nozzle surfaces.

3.      The novelty was not clearly stated in the introduction section. Please rewrite it.

4.      Mistakes:

Line 37: Ra , Ry  should be written as Ra, Ry  

Line 48 Lopez et al.[3] should be written as Lopez et al. [3]

Line 56 Wang et al.[6] should be written as Wang et al. [6]

Line 59 ( Sa and Sq ) should be written as (Sa and Sq)

Line 64 Arnold et al.[8] should be written as Arnold et al. [8]

Please make these corrections in lines 68, 71, 74 and 77

In the whole text, 3D, 2D should be modified as 2-D and 3-D

Line 307 (Figure.7.) Fractal dimension should be written as Figure 7. Fractal dimension

Line 337 Figure.8Fractal dimension should be written as Figure 8. Fractal dimensión

5.      Line 250, In Figure 6, the vertical line on the left side of the figure should be removed.

6.      Figure.9 is not in the text of the article. The number of figures 10, 11 and 12 should be corrected.

7.      There are several typographical and grammatical mistakes which are to be fixed. The abbreviations should be explained on their first appearance, or a separate list of abbreviations may be provided.

8.      The quality of part C of Figures 7 and 8 and the third column of Figures 10 and 11 and Figure 12 should be increased.

9.      In the text of the article, there is no reference to the different values of the fractal dimension. The author can use the following reference Ghobadi, N., Rezaee, S. Synthesis of Ag–Cu–Pd alloy by DC-magnetron sputtering: micromorphology analysis. J Mater Sci: Mater Electron 27, 8464–8477 (2016).

10.   In lines 185 to 200, the inconsistency of the font D, r with the rest of the letters should be corrected.

11.   Please replace the old theoretical articles with newer exprimental articles. Concerning these general subjects, several other representative papers could be cited additionally:

Shahram Solaymani, Ştefan Ţălu, Negin Beryani Nezafat, Sahar Rezaee, Mahboubeh Fathi Kenari. Diamond nanocrystal thin films: Case study on surface texture and power spectral density properties. AIP Advances , 045206 (2020) https://doi.org/10.1063/5.0003866

Author Response

Response to Reviewer 1 Comments

Point 1: Abstract should contain more quantitative information.

Response 1: Thank you for your suggestion . Now I have revised the abstract and the conclusions. And I make sure it is better reflect the content of this article.

Point 2: Introduction may be improved through the inclusion of few more recent previous works done on fuel nozzle surfaces.

Response 2: I have introduced some previous works done on fuel nozzle surfaces and revised the Introduction section.

Point3: The novelty was not clearly stated in the introduction section. Please rewrite it.

Response 3: Thank you for your suggestion . Now I have revised the introduction section. And I make sure it is better reflect the content of this article.

Point 4: Mistakes.

Line 37: Ra , Ry  should be written as Ra, Ry 

Line 48 Lopez et al.[3] should be written as Lopez et al. [3]

Line 56 Wang et al.[6] should be written as Wang et al. [6]

Line 59 ( Sa and Sq ) should be written as (Sa and Sq)

Line 64 Arnold et al.[8] should be written as Arnold et al. [8]

Please make these corrections in lines 68, 71, 74 and 77

In the whole text, 3D, 2D should be modified as 2-D and 3-D

Line 307 (Figure.7.) Fractal dimension should be written as Figure 7. Fractal dimension

Line 337 Figure.8Fractal dimension should be written as Figure 8. Fractal dimensión

Response 4: Thank you for your suggestion .The mistakes have been revised. Figure 7. and Figure 8. have been modified. Both the graphs and fonts must be better.

Point 5: Line 250, In Figure 6, the vertical line on the left side of the figure should be removed

Response 5: The vertical line on the left side of the Figure 6. have been removed.

Point 6: Figure.9 is not in the text of the article. The number of figures 10, 11 and 12 should be corrected

Response 6: Thank you for your suggestion .The number of Figure 8, 9, 10, 11 have been revised.

Point 7: There are several typographical and grammatical mistakes which are to be fixed. The abbreviations should be explained on their first appearance, or a separate list of abbreviations may be provided.

Response 7: The typographical and grammatical mistakes have been revised. The abbreviations have been explained on their first appearance.

Point 8: The quality of part C of Figures 7 and 8 and the third column of Figures 10 and 11 and Figure 12 should be increased.

Response 8: Thank you for your suggestion .The part C of Figure 7. and the Figure 8. have been revised.The third column of Figure 9, 10, 11 also have been modified.

Point 9: In the text of the article, there is no reference to the different values of the fractal dimension. The author can use the following reference Ghobadi, N., Rezaee, S. Synthesis of Ag–Cu–Pd alloy by DC-magnetron sputtering: micromorphology analysis. J Mater Sci: Mater Electron 27, 8464–8477 (2016).

Response 9: Thank you for your suggestion .The new reference have been added.

Point 10: In lines 185 to 200, the inconsistency of the font D, r with the rest of the letters should be corrected.

Response 10: Thank you for your suggestion .The font D,r in the paper have been revised

Point 11: Please replace the old theoretical articles with newer exprimental articles. Concerning these general subjects, several other representative papers could be cited additionally.

Response 11: The old theoretical articles have been replaced by the newer experimental articles. Several other representative papers also be cited.

Reviewer 2 Report

1. The manuscript needs rewriting of several sentences which are ambiguous and doesn't convey clear meaning [see Line 77, 83-84, 86-87, 103, 121, 181, 188, 195, 203, 204 etc.]

2. Authors discussed the previous works in present tense, however it should be in past tense [ see Line 56-64, 72,76, 84 etc.].

3. Line 121- "So, in this paper....[16]". Does it mean the author's manuscript or another previously published work as they have cited one previous work? Sentence required to be rewritten.

4. Line -146: Is it Geometric of surface or geometry of surface? Sentence should be rewritten.

 5. Line 158-159: The sentence required to be rewritten again.

 6. Line 163: Clear and detailed explanation of figures are missing specifically Figure 3 requires good explanation.

7. All Figures are required to be explained clearly.

8. The phrase – “In this paper…” appeared at several place [see line 155, 212, 229, 240]. Why the information about this work is presented in segments at different places? Either the objectives, methodology of the work can be described clearly in the second last paragraph of Introduction as a comprehensive paragraph of information about the work or the sentences required to be reframed.

Author Response

Response to Reviewer 2 Comments

Point 1: The manuscript needs rewriting of several sentences which are ambiguous and doesn't convey clear meaning [see Line 77, 83-84, 86-87, 103, 121, 181, 188, 195, 203, 204 etc.].

Response 1: Thank you for your suggestion . Now I have revised the abstract and the conclusions. And I make sure it is better reflect the content of this article.

Point 2: Authors discussed the previous works in present tense, however it should be in past tense [ see Line 56-64, 72,76, 84 etc.].

Response 2: Now I have revised those sentences in past tense.

Point3: Line 121- "So, in this paper....[16]". Does it mean the author's manuscript or another previously published work as they have cited one previous work? Sentence required to be rewritten.

Response 3: Thank you for your suggestion. I have deleted the “in this paper” .The sentences have been rewritten.

Point 4: Line -146: Is it Geometric of surface or geometry of surface? Sentence should be rewritten.

Response 4: The sentence have been rewriten.

Point 5: Line 158-159: The sentence required to be rewritten again.

Response 5: The sentences have been rewritten.

Point 6: Line 163: Clear and detailed explanation of figures are missing specifically Figure 3 requires good explanation.

Response 6: This part of Figure 3. have been revised.

Point 7: All Figures are required to be explained clearly.

Response 7: All Figures are rewritten clearly.

Point 8: The phrase – “In this paper…” appeared at several place [see line 155, 212, 229, 240]. Why the information about this work is presented in segments at different places? Either the objectives, methodology of the work can be described clearly in the second last paragraph of Introduction as a comprehensive paragraph of information about the work or the sentences required to be reframed.

Response 8: The introduction section have been rewritten and “in this paper” have been modified.

Round 2

Reviewer 1 Report

.......